# Adenosine in Intestinal Epithelial Barrier Function

**DOI:** 10.3390/cells13050381

**Published:** 2024-02-23

**Authors:** Mariya Stepanova, Carol M. Aherne

**Affiliations:** 1Conway Institute of Biomolecular and Biomedical Research, University College Dublin, Belfield, Dublin 4, Ireland; carol.aherne@ucd.ie; 2School of Medicine, University College Dublin, Belfield, Dublin 4, Ireland

**Keywords:** adenosine, IBD, inflammation, mucus, microbiome, tight junctions, chloride secretion, acid–base balance

## Abstract

At the intestinal front, several lines of defense are in place to resist infection and injury, the mucus layer, gut microbiome and strong epithelial junctions, to name a few. Their collaboration creates a resilient barrier. In intestinal disorders, such as inflammatory bowel disease (IBD), barrier function is compromised, which results in rampant inflammation and tissue injury. In response to the destruction, the intestinal epithelium releases adenosine, a small but powerful nucleoside that functions as an alarm signal. Amidst the chaos of inflammation, adenosine aims to restore order. Within the scope of its effects is the ability to regulate intestinal epithelial barrier integrity. This review aims to define the contributions of adenosine to mucus production, microbiome-dependent barrier protection, tight junction dynamics, chloride secretion and acid–base balance to reinforce its importance in the intestinal epithelial barrier.

## 1. Introduction

The intestinal barrier is well equipped to resist luminal injury and pathogen infiltration. Among its defense strategies is the ability to form the mucus layer and strong adhesion junctions [1,2,3,4]. Auxiliary support is also provided by the commensal microbiota, which secrete metabolites that maintain intestinal homeostasis, as reviewed in [5,6,7].

However, the intestinal barrier is not impenetrable. A combination of lifestyle, genetic and environmental factors can weaken the barrier and lend it susceptible to infection and inflammation [8,9,10,11,12]. This chain of events is particularly applicable to inflammatory bowel disease (IBD), which consists of Crohn’s disease (CD) and ulcerative colitis (UC). CD and UC are characterized as chronic relapsing–remitting intestinal disorders. While their exact etiology is unknown, loss of barrier function is thought to be a major contributor to disease onset [13,14,15,16].

Adenosine, a purine nucleoside, acts as a powerful alarm molecule of barrier breakdown. Hypoxia, injury, ischemia and inflammation trigger the release of adenosine triphosphate (ATP) from cells, the canonical adenosine precursor [17,18]. As ATP floods the extracellular compartment, it is rapidly converted to adenosine. The presence of high extracellular adenosine initiates a series of endogenous responses that aim to restore tissue homeostasis.

Adenosine is a retaliatory metabolite that restrains inflammation by direct modulation of immune cell function, as reviewed in [19,20]. This finding has been coupled with complementary studies on its protective effects in intestinal infection and inflammation [21,22,23,24].

However, in the intestinal epithelium, adenosine’s reach extends far beyond the modulation of immune responses to infection and inflammation. It is now appreciated that adenosine is a prominent regulator of intestinal epithelial barrier integrity and repair. While previous reviews have explored the immunomodulatory role of adenosine in intestinal inflammation, this focus has oftentimes obscured adenosine’s more direct role in intestinal barrier function and restoration. The purpose of this review is therefore to discuss the direct effects of adenosine on the individual environments that define the intestinal barrier, so that the full scope of adenosine’s contributions to intestinal homeostasis may be fully appreciated. As such, adenosine production, regulation and signaling will be addressed; subsequent sections will delve into adenosine’s distinct relationship with mucus production, microbiome-dependent barrier protection, tight junction dynamics, chloride secretion and acid–base balance.

## 2. The Adenosine Alarm System

### 2.1. Adenosine Production

As outlined above, adenosine is derived from extracellular or intracellular ATP. The sequential degradation of extracellular ATP is achieved by two extracellular membrane-bound enzymes: CD39 (ectonucleoside triphosphate diphosphohydrolase 1, E-NTPDase1) and CD73 (ecto-5′-nucleotidase, Ecto 5′NTase). CD39 cleaves extracellular ATP to adenosine diphosphate (ADP) and then to adenosine monophosphate (AMP) [25,26,27]. CD73 cleaves the resulting AMP into adenosine [25,27,28,29] (Figure 1). In an alternative pathway, extracellular ATP can be directly broken down to AMP by the ecto-nucleotide pyrophosphatase/phosphodiesterase family member 1, CD203a, and the resulting AMP can then be converted to adenosine by CD73 [25,29,30]. Adenylate kinase-1 (AK1) regulates phosphotransfer reactions and can remove a phosphoryl group from ADP to generate AMP, allowing it to be converted to adenosine by CD73 [25,31,32,33] (Figure 1).

It is important to note that the accumulation of adenosine in the extracellular environment is not solely attributable to the presence of extracellular ATP. In the non-canonical CD38/CD203a/CD73 axis, the CD38 enzyme can catalyze the conversion of nicotinamide adenine dinucleotide (NAD+) to adenosine diphosphate ribose (ADPR), which can also be converted to AMP by CD203a and then finally to adenosine by CD73 [25,30,34] (Figure 1). Adenosine can also form in the intracellular compartment. In cell bioenergetics, the hydrolysis of ATP generates ADP, which itself can be further hydrolyzed to AMP. In this cascade, AMP can be cleaved into adenosine through the actions of cytosolic 5′-nucleotidase (cyto-5′NT) [25,35]. In an alternative pathway, S-adenosylhomocysteine hydrolase (SAHH) can hydrolyze S-adenosylhomocysteine (SAH) into adenosine and homocysteine [25,36,37] (Figure 1).

### 2.2. Modulation of Adenosine Levels 

Shuttling of adenosine across the plasma membrane is facilitated by two classes of solute carriers (SLCs): SLC28 and SLC29. The SLC28 family encompasses three human concentrative nucleoside transporters (CNTs), CNT1, CNT2 and CNT3, which transport various nucleosides into the intracellular compartment by coupling to the influx of sodium ions (Na^+^), and in the case of CNT3, also hydrogen ions (H^+^) [38,39,40,41,42,43,44,45] (Figure 2). While CNTs display selectivity to a profile of purine and pyrimidine nucleosides, only CNT2 and CNT3 allow adenosine translocation [41,45,46] (Figure 2). In polarized epithelial cells, CNTs preferentially localize to the apical surface [46,47,48]. In colonic epithelial cells, this retention is facilitated by galectin-4, which anchors CNT3 to the apical surface [49].

The SLC29 family represents four human equilibrative nucleoside transporters (ENTs): ENT1, ENT2, ENT3 and ENT4. In contrast, these shuttle nucleosides and nucleobases in a bidirectional manner across the membrane based on their concentration gradients [45,50] (Figure 2). ENTs display a broad permeability to nucleosides and adenosine transport at the plasma membrane occurs through ENT1, ENT2 and ENT4 (alias: plasma membrane monoamine transporter (PMAT)) [45,51,52,53,54,55]. ENT3 also transports adenosine, however it is localized on intracellular membranes of the mitochondria and lysosomes, which precludes its ability to regulate extracellular adenosine signaling events during inflammation and injury [45,56,57]. ENT1 and ENT2 have been suggested to localize to the basolateral membrane of polarized cells, however, their apical expression has also been reported [47,50,58,59,60]. Meanwhile, ENT4 appears to localize to the apical surface [61]. An acidic pH increases adenosine uptake by ENT1, ENT2 and ENT4, which implicates the ENTs’ involvement in acidotic conditions such as ischemia and inflammation [55,61,62,63]. 

Counterbalancing excessive adenosine production is adenosine deaminase (AD). Acting at both the intracellular and extracellular front, AD irreversibly deaminates adenosine into inosine, a precursor to uric acid [25,64,65] (Figure 1). The combined contributions from SLCs and AD fine-tune adenosine levels in the extracellular space and control the initiation and duration of the adenosine alarm response at the intestinal front.

### 2.3. Adenosine Receptors

In the extracellular environment, adenosine couples to adenosine receptors, a family of G-protein coupled receptors (GPCRs). Four subtypes of adenosine receptors are recognized: A1 adenosine receptor (A1AR), A2A adenosine receptor (A2AAR), A2B adenosine receptor (A2BAR) and A3 adenosine receptor (A3AR) [66,67]. The subdivision of the two A2 receptors into A2AAR and A2BAR reflects their different binding affinities to adenosine, with A2BAR having a lower affinity for adenosine relative to A2AAR [67]. In fact, A2BAR possesses the lowest affinity for adenosine out of all the receptor subtypes and is activated solely in circumstances of high extracellular adenosine, such as during inflammation and injury [67,68]. This defining feature of A2BAR often makes it the critical responder through which adenosine elicits its protective effects [69]. In addition, A2BAR is abundantly expressed on the apical and basolateral membranes of intestinal epithelial cells, particularly those of the colon [22,70,71]. Considering its activation threshold and extensive expression on the surface of intestinal epithelial cells, the impact of A2BAR signaling has gathered particular interest in intestinal homeostasis [21,22,24,72].

### 2.4. Adenosine Receptor Signaling

Owing to their nature as GPCRs, adenosine receptors couple to various G-protein α-subunits to initiate distinct signaling events. Broadly speaking, the A2 adenosine receptors, A2BAR or A2AAR, couple to the Gs α-subunit (Gs), which stimulates adenylyl cyclase (AC) to produce cyclic-AMP (cAMP) [67,68,73,74] (Figure 3). The availability of intracellular cAMP activates protein kinase A (PKA) to initiate PKA-dependent signaling cascades and transcription events. 

Conversely, A1AR and A3AR interact with the Gi α-subunit (Gi) and Gq α-subunit (Gq). In this setting, coupling of A1AR and A3AR to Gi inhibits adenylyl cyclase activity, cAMP formation and PKA signaling [68,75,76] (Figure 3). Meanwhile, A1AR and A3AR signaling through Gq activates phospholipase C (PLC), which in turn increases the levels of inositol 1,4,5-triphosphate (IP3)/diacylglycerol (DAG) [76,77,78,79]. DAG proceeds to activate protein kinase C (PKC), while IP3 increases intracellular concentration of calcium (Ca^2+)^ by promoting its release from the endoplasmic reticulum (ER) [77,78,79,80] (Figure 3). 

While not the focus of this review, it is worth mentioning that adenosine receptor coupling to G-proteins is tissue and context dependent and not restricted to the pathways mentioned above [81,82,83,84]. Adenosine receptor signaling through ligand-independent mechanisms has also been reported [85,86]. Collectively, it has provided us with an appreciation for the diverse ways through which adenosine initiates intracellular signaling. Subsequent sections of this review explore the identified adenosine-mediated pathways, from the receptors to the intracellular signaling cascades that impact intestinal barrier function.

## 3. Adenosine in the Intestinal Mucus Layer

The surface of the intestinal epithelium is lined by a sheath of mucus, which acts as the frontline layer of protection. The small intestine is protected by a single layer of loose mucus, while the colon is lined by a mucus bilayer [3,87,88]. The production and maintenance of the mucus layer is assigned to goblet cells, a specialized lineage of intestinal epithelial cells. Goblet cells constitutively secrete mucins, the main components of the mucus layer [89]. Mucins are categorized as either membrane-associated mucins or secreted mucins. Secreted mucins are further subdivided as either gel-forming mucins, which become hydrated and extensively linked to each other on the surface of the epithelium, or non-gel forming mucins, whose role is poorly defined. In the colon, the dominant mucin is the secreted gel-forming mucin MUC2/Muc2 [3]. Murine deletion of *Muc2* results in bacterial contact with the colonic epithelium, infection, development and exacerbation of colitis, as well as onset of colorectal cancer [3,4,13,90,91,92]. Prior to secretion, Muc2 is stabilized by sialylation by ST6 sialyltransferase [93]. This represents a key step in Muc2 processing to resist degradation by bacterial enzymes [93]. The stability of Muc2 is of critical importance to mucus formation. Mice harboring a ST6 sialyltransferase mutation seen in IBD patients have more severe dextran sulphate sodium (DSS)-induced colitis compared to the wildtype control [93].

Among its diverse roles, adenosine has been shown to regulate mechanisms of mucus production and secretion, particularly in the pulmonary epithelium. To date, limited evidence exists on adenosine’s involvement in intestinal mucus production. It has been suggested that adenosine signaling through the A2B adenosine receptor (A2BAR) promotes Muc2 secretion and the restoration of the mucus layer [94]. Following helminth infection, mice deficient in *A2BAR* (*A2BAR*^−/−^) exhibited a reduction in goblet cells and an associated reduction in *Muc2* expression in their small intestine, compared to the wild-type control [94]. However, in murine colitis, no significant difference in *Muc2* expression was observed in the colon of *A2BAR*^−/−^ mice, or in mice with a tissue-specific deletion of *A2BAR* (*Adora2b^fl/f^*^l^
*VillinCre*^+^), compared to their respective controls [21]. The lack of consensus between the two studies may stem from different models of intestinal dysfunction, the fact that Muc2 was measured in different segments of the intestine, or that the studies varied between examining the effects of global or tissue-specific deletion of *A2BAR*.

Insight can be drawn from adenosine’s regulation of pulmonary mucus. In chronic airway diseases, such as asthma and chronic obstructive pulmonary disease (COPD), adenosine accumulation promotes mucus secretion and hyperplasia through A1AR- and A3AR-mediated signaling [95,96,97]. Unfortunately, in these studies, adenosine-driven mucus overproduction contributes to airway obstruction and exacerbates disease. However, in alternative studies, double knockout of *A2AAR*/*AD* or *A1AR*/*AD* in mice results in mucus metaplasia and exaggerated pulmonary inflammation, suggesting that adenosine signaling dampens pulmonary mucus production [98,99]. 

These conflicting reports on both adenosine-driven hypersecretion and restraint of mucus production in both intestine and lung reflect the complexity of adenosine signaling and may suggest disease model-specific or receptor-specific effects. Further studies are needed to tease out the possible role(s) for adenosine in intestinal mucus production. Clarifying this relationship would be of considerable benefit in intestinal diseases, such as IBD where mucus layer formation is notoriously impaired [4,13,14] (Figure 4).

## 4. Contributions of the Gut Microbiome to Adenosine-Mediated Barrier Protection

The intestinal barrier is a hub for commensal bacteria. Collectively referred to as the microbiome, or microflora, they extensively colonize the outer mucus layer of the colon [3]. This permeable and glycan-rich environment provides protection and can act as a source of sugar that sustains their growth [100,101,102]. In return, the microbiome repays the host’s hospitality by providing colonization resistance against pathogenic bacteria [103,104,105]. 

The contributions of the microbiome to barrier function extend beyond their prevention of pathogen infiltration. The secretion of microbiome-derived metabolites creates an intricate communication network between the microbiome and the intestinal epithelium. The impact of microbial-derived metabolites on intestinal homeostasis and IBD is reviewed elsewhere [5,6,7,106]. Among the diverse roles of the microbiota is the ability to modulate host immune responses and reinforce barrier integrity [107,108,109,110,111,112,113]. It goes without saying that alterations in the composition of the gut microbiome and in the levels of microbiome-derived metabolites are associated with IBD [114,115,116,117,118]. In the landscape of adenosine signaling, the intestinal microbiota is also known to secrete the adenosine precursor, ATP [119,120,121]. Meanwhile, *Lactobacillus reuteri* treatment has been shown to induce expansion of beneficial gut microbiota and augment the plasma levels of adenosine in mice [122] (Table 1). Taking this into account, an innovative approach is to administer engineered probiotics. Scott et al. have devised self-tunable extracellular ATP (eATP)-responsive yeasts, capable of secreting CD39 in a time- and location-specific manner upon detection of extracellular ATP to stimulate its conversion to the anti-inflammatory adenosine [123]. In their murine models of colitis and enteritis, treatment with eATP-responsive yeasts resulted in a reduction in intestinal inflammation, fibrosis and gut dysbiosis, which propels adenosine-producing biotherapeutics as attractive novel therapies in IBD [123]. 

Another facet of adenosine’s function is its direct antimicrobial effect. In vitro evidence shows that adenosine has a potent bacteriostatic effect on the growth of *Salmonella enterica* serovar *Typhimurium* [124]. While in mice with an intestinal epithelial-specific deletion of *CD73* (*CD73^f/f^ VillinCre*), *Salmonella* colonization was greatly increased compared to the Cre-negative control mice [124]. Taken together, it suggests that CD73 activity, and by extension adenosine production, suppress *Salmonella* colonization. The study does elaborate that the absence of *CD73* expression on the intestinal epithelium of *CD73^f/f^ VillinCre* mice greatly reduces *Salmonella* virulence, colitis and dissemination, implying that CD73 function is required for *Salmonella* infection [124] (Table 1). However, the exact mechanism that increases *Salmonella* virulence is yet to be determined. More recently, it was outlined that the bacteriostatic effect of adenosine may stem from its ability to dysregulate bacterial metabolism and enhance bacterial killing when administered in conjunction with antibiotics [125]. 

**Table 1 cells-13-00381-t001:** Summary of the relationship of commensal and pathogenic bacteria with adenosine.

Bacteria	Regulates Adenosine Signaling	Affected by Adenosine Signaling	Effect	Reference
**Commensal Bacteria:**				
*Lactobacillus reuteri*	✓		Increases plasma levels of adenosine	[122]
*Bifidobacterium pseudolongum*	✓		Capable of secreting inosine, a derivative of adenosine that is known to activate the A2A adenosine receptor and protect the intestinal epithelium during colitis.	[126,127,128,129]
*Akkermansia muciniphila*	✓		Capable of secreting inosine, a derivative of adenosine that is known to activate the A2A adenosine receptor and protect the intestinal epithelium during colitis.	[126,127,128,129]
**Pathogenic Bacteria**				
*Salmonella enterica* (*serovar Typhimurium*)		✓	Adenosine induces a bacteriostatic effect CD73 activity suppresses colonization but increases virulence	[124]

Interestingly, there also exists another metabolite known to regulate adenosine signaling and barrier function. This metabolite is inosine, the product of adenosine deamination. Microbiota *Bifidobacterium pseudolongum* and *Akkermansia muciniphila* have been shown to contribute to intestinal inosine production [126] (Table 1). In *Bifidobacterium pseudolongum*–monocolonized mice, the production of inosine appears to exist in a gradient, with high levels in the duodenum, followed by a gradual decrease along the jejunum and caecum [126]. Inosine has been shown to directly bind and activate adenosine receptors and is recognized for its ability to produce an anti-inflammatory response through A1AR, A2AAR and A3AR signaling [127,130,131,132,133,134]. In murine DSS-induced colitis, inosine attenuates the hallmarks of colitis and colonic levels of malondialdehyde (MDA), myeloperoxidase activity (MPO), major intrinsic proteins (MIP)-1α and -2 and proinflammatory cytokines IL-1, IL-6, IL-12 and TNF-α [128]. The involvement of inosine-A2AAR signaling in colitis was subsequently suggested. In trinitrobenzenesulfonic acid (TNBS)-induced colitis in rats, inosine was shown to reduce macroscopic injury, as well as levels of MDA, MPO, IL-1β and TNF-α in colonic tissue, in part, through the activation of A2AAR [129]. Since then, it has been elaborated that microbiota-derived inosine attenuates colitis and ameliorates colonic motility through A2AAR-dependent activation of peroxisome proliferator-activated receptor-γ (PPARγ) signaling [127]. 

Taking the current knowledge together, it appears the microbiome may be a source of adenosine, which can have direct protective effects on the host as well as an ability to modulate functional responses of the microbiota. The convergence of microbiome-derived inosine (an adenosine metabolite) with adenosine receptor signaling broadens our understanding of adenosine as an active participant in intestinal barrier homeostasis. It would be of interest to confirm whether microbiota-derived adenosine provides similar A2AAR-dependent protection of barrier function in colitis like its close counterpart inosine. The emerging evidence on the ability of adenosine to regulate bacterial metabolism is intriguing, especially in the context of antibiotic killing. Further studies are needed to explore the possible interplay between adenosine-mediated effects on the host and the resident microbiota in the context of homeostasis and disease.

## 5. Adenosine Reinforces Epithelial Tight Junctions

### 5.1. Tight Junction Architecture

The paracellular space between adjacent intestinal epithelial cells is firmly sealed by junctional complexes. The junctional complexes are located at the apical regions of the lateral membranes of cells and consist of tight junctions, adherens junctions and desmosomes [135,136]. The contributions of adherens junctions [137,138,139] and desmosomes to barrier function are well recognized [140,141,142,143,144]. However, tight junctions (TJs) act as the critical determinants of intestinal epithelial barrier integrity [135]. A more detailed description of TJs structure, function and association with intestinal disease has been well reviewed elsewhere [135,136,145,146]. Briefly, several TJ fusion sites are required to firmly join the lateral membranes of opposing cells to each other and collectively, they are known as the TJ strand [136]. The TJ strand prevents pathogen infiltration through the paracellular space while regulating the movement of molecules across the membrane [136,145,146]. The enduring presence of tight junctions along the span of the intestinal epithelium also regulates apical and basolateral polarity [145,146]. Claudins and zonula occludens-1 (ZO-1) represent core components of TJ architecture, with reinforcement provided from occludins, tricellulins, MarvelD3 and junctional adhesion molecules (JAMs) [135,145]. 

TJs are fluid and dynamic in nature. Their assembly and disassembly is heavily influenced by a wide variety of stimuli, such as calcium, cytokines, immune responses and bacterial presence [147,148,149,150,151,152]. IBD, in particular, is a notorious culprit of TJ restructuring and dissociation [16,153,154] (Figure 4).

### 5.2. CD39 and CD73 Contributions to Barrier Function

Adenosine is assigned the task of preserving barrier integrity in both homeostasis and disease. In infection, inflammation and hypoxia, adenosine signaling is responsible for the initiation of repair pathways and TJs re-formation. By extension, adenosine production through the canonical CD39–CD73 axis acts as cornerstone of barrier restitution. CD39–CD73-driven adenosine production is required for the maintenance of both vascular and intestinal barrier integrity. 

In the vascular barrier, loss of *CD39* and *CD73* results in diminished adenosine production and increased vascular permeability [155,156]. In hypoxia, CD39 and CD73 activity prevents vascular leakage, potentially via adenosine-mediated activation of A2BAR on the endothelial cells [157,158]. 

The importance of CD39 and CD73 is echoed in the intestinal epithelium. Murine loss of *CD39* contributes to increased susceptibility to colitis [159]. Polymorphisms in *CD39* are also strongly associated with Crohn’s disease [159]. Meanwhile, murine deficiency in *CD73* results in increased intestinal permeability, downregulation of epithelial tight junction proteins JAM-A and claudin-2 and adherens junction protein α-catenin, and also leads to increased susceptibility to colitis [160]. In another study, inhibition of CD73 on intestinal epithelial cells led to an increase in barrier permeability upon exposure to *Clostridium difficile* (*Cdf*) toxins, which was potentially attributable to dysregulation of ZO-1 localization [161]. Supporting in vivo work confirmed that inhibition of CD73 in mice increases colonic damage and epithelial permeability after exposure to *Cdf* toxins [161]. In intestinal hypoxia, CD39 and CD73 expression and activity is upregulated as a retaliatory response from the intestinal epithelial cells. While in vivo inhibition of CD73 results in increased intestinal permeability in both hypoxic and control mice [162]. Taken together, the overarching consensus from these studies is that the adenosine-producing ectoenzymes, CD39 and CD73, play a critical role in intestinal epithelial barrier integrity.

### 5.3. Adenosine Receptor Signaling in Tight Junction Dynamics

As described above, adenosine-producing enzymes play a regulatory role in intestinal epithelial barrier function and junction organization. Initially, it was suggested that the protective effect can be, in part, attributed to adenosine signaling through the A2AAR and A2BAR. Evidence taken from the vascular barrier reveals that NECA, an adenosine analogue, and non-selective agonist of A2AAR and A2BAR, promotes PKA activation and downstream phosphorylation of vasodilator-stimulated phosphoprotein (VASP)—which strengthens vascular barrier by co-localizing with ZO-1, occludin and JAM-1 at the endothelial junction [163]. This adenosine–PKA–VASP axis was shown to be conserved in the intestinal epithelium, where NECA also stimulates PKA-dependent phosphorylation of VASP and promotes its co-localization with ZO-1 at the junctional border to restore barrier integrity [164].

NECA does not discriminate between A2AAR and A2BAR and acts as a non-selective agonist of both receptors. Clarification on which A2 adenosine receptor subtype was responsible for the intestinal epithelial barrier protection came later in studies using A2BAR knockout (*A2BAR*^−/−^) mice. Here, it was shown that *A2BAR*^−/−^ mice have heightened susceptibility to DSS-induced colitis [22]. Pharmacological inhibition of A2BAR also increases the severity of DSS-induced colitis in wildtype controls [22]. The protective effects of A2BAR on intestinal function were reiterated in models of murine ischemia/reperfusion (IR) injury, where *A2BAR*^−/−^ mice exhibited increased injury, compared to the wildtype control mice [165]. Subsequent treatment with the A2BAR selective agonist, BAY 60-6583, attenuated IR injury in wildtype mice but not in *A2BAR*^−/−^ mice [165]. Wildtype mice treated with BAY 60-6583 also demonstrated decreased intestinal permeability, which directly implicated A2BAR signaling in barrier protection [165]. The specific mechanism by which A2BAR mediates its barrier protective effects was later proposed using mice with an intestinal epithelial-specific deletion of *A2BAR* (*Adora2b^fl/fl^ VillinCre*^+^) [21]. In DSS-colitis, *Adora2b^fl/fl^ VillinCre*^+^ mice showed increased disease severity coupled with increased intestinal permeability [21]. This was supported by in vitro observations that knockdown of *A2BAR* in intestinal epithelial cells significantly reduced the rate of barrier resealing following calcium switch [21]. It was clarified that the protective effects of A2BAR on barrier function stem from PKA-dependent phosphorylation of VASP in the intestinal epithelium, which echoes previous findings. However, in this model of colitis, phospho-VASP coordinated barrier restitution by co-localization with E-cadherin, an integral protein of adherens junctions [21] (Figure 5).

It becomes of relevance to highlight that tight junctions and adherens junctions are known to share ZO-1 as a common interactor and ZO-1 is capable of co-localizing with E-cadherin [166,167,168,169,170]. It is plausible that phospho–VASP co-localization with E-cadherin at the adherens junction is observed because of its interactions with ZO-1, as described previously. Clarifying whether phospho–VASP directly engages ZO-1 and whether this interaction regulates tight junction or adherens junction formation would be of interest. In this branch, the role of the adenosine–PKA–VASP axis in the recruitment of tight junction proteins (claudins, occludins and JAMs) and adherens junction proteins (cadherins, catenins) remains to be explored. Desmosome dysregulation is also implicated in IBD and their contributions to barrier function are not negligible [140,141,142,143,144,171]. Addressing the interplay of adenosine signaling with desmosome formation would resolve the knowledge gap in the field. 

Conflicting reports demonstrate a detrimental role of A2BAR in the intestine. In alternative studies, pharmacological inhibition of A2BAR and *A2BAR* genetic deletion result in a decrease in the severity of DSS-colitis in mice [172,173]. Furthermore, *A2BAR*^−/−^ mice were more resistant to TNBS-induced colitis and *Salmonella Typhimurium*–induced colitis, but not systemic *Salmonella Typhimurium* sepsis [172]. Using bone-marrow chimeras, Ingersoll et al. reported that murine DSS-induced colitis was specifically attenuated in the absence of mucosal-*A2BAR* [174]. In a novel therapeutic approach, they also reported that treatment with *A2BAR*-siRNA loaded nanoparticles decreased the production of proinflammatory cytokines and improved DSS-colitis [174]. In hypoxia-induced intestinal permeability, inhibition of A2BAR increased claudin-1, occludin and ZO-1 mRNA and protein expression in vitro [175]. Similarly, in murine IR injury, A2BAR antagonism reduced intestinal permeability and increased claudin-1, occludin and ZO-1 mRNA and protein expression in vivo [175], implying a detrimental role of A2BAR activity on TJ restoration. 

The conflicting reports on A2BAR’s role in intestinal barrier function in vivo may stem from the genetic background of the mice used, and their housing environments, which may have resulted in diverging microbial and immune profiles, leading to different outcomes in disease models. The tissue-specific contributions of adenosine receptors cannot be underestimated in this context. Indeed, whole body deletion of *A2BAR* in *Salmonella*-induced colitis was protective, but exacerbated *Salmonella*-induced sepsis, which suggests possible cell- and tissue-specific effects of A2BAR signaling. This is further supported by studies in which specific deletion of *A2BAR* in the vasculature did not affect the outcome of DSS-colitis but specific deletion of *A2BAR* in the intestinal epithelium was demonstrated to be barrier protective in the same model [21]. In the context of tight junction dynamics, studies in cells and mice with an *A2BAR* deletion and agonist treatment have consistently pointed to a link between A2BAR and mechanisms supporting tight junction function, in both the vascular and intestinal epithelial barriers. Future studies should thus veer towards using a tissue-specific knockout of *A2BAR* to eliminate confounding variables and clarify the extent of A2BAR’s involvement in barrier protection. 

Another point of consideration is the possible role of A2AAR. This may be of relevance in the context of *A2BAR* deletion or antagonism, where extracellular adenosine is still readily available and may signal through A2AAR instead. To date, the role of A2AAR in intestinal barrier function has been poorly described; however, given the parallels in intracellular signaling between the two receptors, it may perform a similar role to A2BAR.

## 6. Adenosine Promotes Intestinal Chloride Secretion

The intestinal epithelium is tasked with regulating electrolyte transport [176]. The specific electrolyte, chloride (Cl^−^), acts as a key determinant of mucus layer hydration and fluid balance [176,177]. Chloride accumulates inside epithelial cells through the actions of the NKCC1 cotransporter located on the basolateral side. NKCC1 shuttles sodium, potassium and chloride inside the cells in a 1:1:2 ratio. Importantly, chloride accumulates inside the cells above its electrochemical equilibrium and the opening of Cl^−^ efflux channels allow for Cl^−^ to freely move out into the lumen [177]. In the colon, Cl^−^ efflux occurs through the cystic fibrosis transmembrane conductance regulator (CFTR) or through Ca^2+^ dependent Cl^−^ channels [178,179]. As Cl^−^ ions move out of the cell, they create an osmotic gradient that promotes water movement into the lumen to support mucus layer hydration [177]. At this interface, adenosine has been well recognized to promote Cl^−^ secretion in intestinal epithelial cells [180,181,182,183]. Adenosine signaling through A2BAR initiates cAMP-dependent Cl^−^ secretion in intestinal epithelial cells [71] (Figure 5). In vivo studies in mouse jejunum also reveal that, apical, but not basolateral stimulation of A1AR promotes Cl^−^ secretion [184]. The same study additionally reports that A2A signaling has no impact on the regulation of chloride secretion [184].

While Cl^−^ is required for mucus layer hydration via its osmotic regulation of water movement, excessive Cl^−^ is the cause of secretory diarrhea [177,179]. Suffice to say, it has not been clarified whether adenosine-driven chloride secretion directly impacts mucus layer hydration. However, its activity does not appear to result in Cl^−^-induced diarrhea. In a study evaluating the loss of *A2BAR* expression on mice with DSS-induced colitis, Frick et al. reported an increase in the disease index activity for *A2BAR* deficient mice, with one of the parameters being stool consistency [22]. It can be implied that stool consistency was therefore not considerably impacted in mice with preserved A2BAR expression during DSS-induced colitis. The caveat of this assumption is that stool consistency was assessed alongside multiple other parameters to give a global score of disease severity. The scoring of stool consistency specifically was not addressed by the authors. In this regard, reviewing existing in vivo data and teasing out the impact of adenosine signaling on stool consistency may be a worthwhile expansion to our understanding of adenosine-driven Cl^−^ secretion.

## 7. Adenosine Restores Intestinal Acid–Base Balance

Acidification is a recognized feature of intestinal inflammation [185,186,187,188]. The shift in pH towards acidification results from the accumulation of immune cells at the inflamed site and increased release of lactate from intestinal epithelial cells [188,189]. The ability to maintain pH homeostasis requires intestinal epithelial-mediated secretion of bicarbonate (HCO_3_^−^) into the lumen. The inability to resolve lactate-induced acidity and increase HCO_3_^−^ secretion is associated with murine colitis, CD and UC [186,189,190,191].

The major promoter of HCO_3_^−^ secretion in the colonic, ileal, duodenal epithelium is the chloride anion transporter, SLC26A3 [189,192,193,194,195,196,197]. SLC26A3 is a key regulator of acid–base homeostasis that facilitates Cl^−^ absorption and HCO_3_- secretion. Its dual role enables it to mitigate Cl^−^ induced diarrhea while restoring intestinal pH through HCO_3_^−^-driven alkalization [189,197,198]. Indeed, patients with UC and CD exhibit markedly reduced SLC26A3 expression [189,199,200]. It has also been elaborated that SLC26A3 significantly contributes to intestinal epithelial barrier function. In vivo knockdown of *SLC26A3* in mice resulted in decreased Muc2 staining and impaired mucus layer formation [201]. While in vitro knockdown of *SLC26A3* in intestinal epithelial cells distorted the localization of tight junction proteins occludin, claudins and ZO-1 [202]. In the same study, *SLC26A3* overexpression using intracolonic delivery of an adenovirus harboring the *SLC26A3* gene protected mice from TNF-induced disruption of TJ proteins [202]. Previous studies have implicated the cAMP signaling pathway in the regulation of *SCL26A3* [193,203]. In line with this evidence, adenosine has been shown to induce *SLC26A3* expression through the cAMP-CREB pathway and limit intestinal acidification [189].

Taken together, loss of SLC26A3 function results in the excessive accumulation of Cl^−^ and absence of HCO_3_^−^ alkalization. This two-pronged effect has profound consequences on barrier integrity. Distortions in mucus layer formation, tight junction architecture coupled with diarrhea and intestinal acidification all contribute to the development and exacerbation of colitis. At this front, the ability of adenosine to simultaneously regulate both Cl^−^ efflux and influx points to the possible existence of an equilibrium that is required for adenosine-mediated barrier protection. It may prove with time that the protective effects of adenosine stem from a loop mechanism, in which adenosine initially promotes Cl^−^ efflux for mucus hydration before stimulating SLC26A3-driven uptake of Cl^−^ to maintain intestinal barrier function. In IBD, this adenosine-driven loop mechanism may serve to reset intestinal homeostasis.

## 8. Contribution of ENTs to Intestinal Barrier Function

As we have outlined, there is a complex network of enzymes and transporters that fine-tune the levels of extracellular adenosine. In compiling this review, it was evident that studies to date have focused on the enzymes that regulate adenosine production and on adenosine receptor signaling. Limited information exists on the potential role of adenosine transporters in the regulation of intestinal barrier function. CNTs that shuttle adenosine against a concentration gradient have been demonstrated to be expressed on intestinal epithelial cells and *CNT2* expression is elevated in the colon of IBD patients [204]. Therefore, while *CNT* expression appears to be dysregulated in the injured intestine it is unknown if this affects adenosine concentrations at the epithelial surface and if this might have a functional consequence for intestinal barrier function.

More is known about the ENTs, which regulate the duration of the adenosine response at the intestinal interface and shuttle adenosine in a bidirectional manner across the plasma membrane according to its concentration gradient. To date, *ENT1* and *ENT2* have been identified to be expressed in the intestine [24,204]. The exact expression pattern of ENTs in the intestine is somewhat in doubt as one study failed to identify substantial expression of *ENT1* in normal colonic tissue [204]. However, the same study demonstrated an increase in *ENT1* and *ENT2* expression in the inflamed ileum and colon of IBD patients [204]. In other studies, *ENT1* and *ENT2* have been demonstrated to be expressed in the intestinal epithelium and reduced expression of *ENT2* was observed in IBD biopsies, murine colitis [24] and in hypoxia [59]. 

Functional studies have been undertaken to examine a possible role of ENT2 in the intestine. Mice with global knockout of *Ent2* (*Ent*^−/−^) and intestinal epithelial-specific knockout of *Ent2* (*Ent2^fl/fl^VillinCre*^+^) have allowed us to better understand the impact of prolonged adenosine signaling. Both *Ent2*^−/−^ and *Ent2^fl/fl^VillinCre*^+^ mice were protected during DSS-induced colitis and showed a reduction in intestinal permeability [24]. It was subsequently confirmed in *Ent2*^−/−^ mice and with Ent2 pharmacologic inhibition that these protective effects stemmed from an increase in the extracellular adenosine levels in the colon, which activated A2BAR signaling events [24]. 

Collectively, this evidence points to a protective response, where the decrease in *ENT2* expression in IBD patients may reflect enhanced adenosine accumulation and initiation of A2BAR signaling to alleviate intestinal epithelial damage and reduce barrier permeability [24]. However, given the conflicting findings related to ENT expression in disease and the limited functional studies available [24,204], more work is needed to define the potential role of ENTs at the intestinal barrier.

## 9. Concluding Remarks and Future Perspectives

Adenosine’s entanglement with gut microbiota, mucus production, tight junctions, chloride secretion and intestinal pH highlights the extent of its protective effects on intestinal barrier function. Coordinating these protective effects are not only the adenosine receptors but also adenosine-producing enzymes and adenosine transporters. This large collaborative network modulates the duration of the adenosine response in the extracellular compartment, with the aim of strengthening intestinal barrier integrity. When superimposed with adenosine’s previously established role in dampening inflammatory responses, adenosine signaling becomes of particular importance in diseases of the intestinal barrier, such as IBD, where it may act as a potential therapeutic target. 

Indeed, targeting intestinal epithelial restitution has gained increased traction in the field of IBD therapeutics, given the pitfalls of the current anti-inflammatory therapies. At the intestinal interface, barrier integrity is governed by host–bacterial interactions coupled with mucus production. Intriguing evidence suggests adenosine could contribute to mucus production and control host–bacterial interactions. Expanding on these findings would complement the existing evidence on the protective effects of adenosine during intestinal barrier disruption. The most substantial amount of work has focused on adenosine as a molecule capable of strengthening the intestinal barrier through tight junction regulation. However, in barrier dysfunction observed during IBD, increased intestinal permeability is also associated with the disruption of adherens junctions and desmosomes. Restoration of barrier integrity requires co-ordination between all junctional complexes. Defining the impact of adenosine on adherens junctions and desmosome dynamics would complete our understanding of adenosine’s role in intestinal epithelial barrier integrity. 

Many studies fall short in establishing the specific mechanism through which adenosine receptors mediate their effects. While A2BAR appears to be the primary receptor through which adenosine protects intestinal epithelial barrier function, A2BAR activation is contingent on high levels of circulating adenosine, which is present only during pathological states, such as IBD. Addressing how A1AR, A2AAR and A3AR contribute to intestinal barrier function, when adenosine levels are below the threshold of A2BAR activation, is therefore a key future direction. Furthermore, there is a need to go beyond receptors and identify the downstream molecules involved in adenosine receptor protective signaling events, which to date are largely unknown.

Adenosine is a multifaceted molecule, and we are still far from understanding the full scope of its effects on the intestinal epithelial barrier. Further studies addressing the unanswered questions we highlight would reveal a more complete picture of adenosine contributions at the intestinal front. The creation of self-tunable adenosine-producing biotherapeutics, such as the eATP-responsive yeasts that alleviate the severity of colitis and enteritis is an exciting new development in the field that may serve as a potential future therapeutic strategy during intestinal barrier disruption. Taken together, the existing evidence positions adenosine as a decisive regulator of intestinal epithelial barrier integrity with possible therapeutic potential.

## Figures and Tables

**Figure 1 cells-13-00381-f001:**
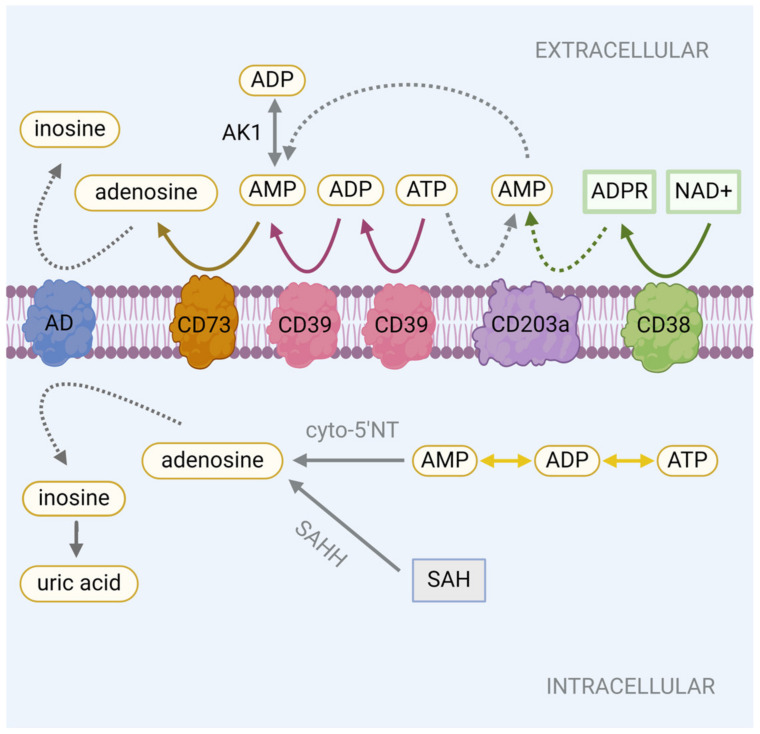
The diverse mechanisms of adenosine production. Abbreviations: adenosine triphosphate (ATP), adenosine diphosphate (ADP), adenosine monophosphate (AMP), adenylate kinase-1 (AK1), adenosine deaminase (AD), nicotinamide adenine dinucleotide (NAD+), adenosine diphosphate ribose (ADPR), cytosolic 5′-nucleotidase (cyto-5′NT), S-adenosylhomocysteine hydrolase (SAHH), S-adenosylhomocysteine (SAH). Created with BioRender.com, accessed on 13 February 2024.

**Figure 2 cells-13-00381-f002:**
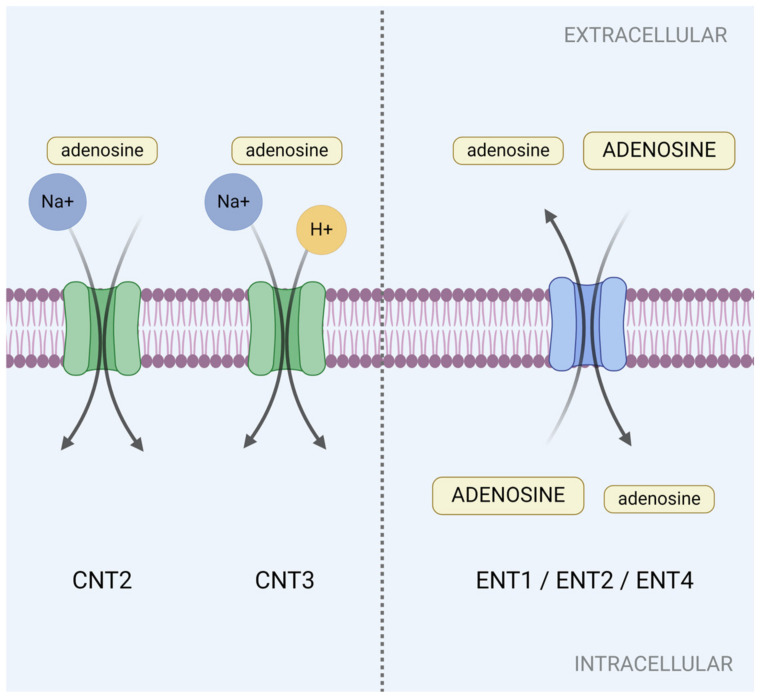
Regulation of adenosine levels across the plasma membrane is facilitated by concentrative nucleoside transporters (CNTs): CNT2 and CNT3 and equilibrative nucleoside transporters (ENTs): ENT1, ENT2, ENT4. CNTs transport adenosine into the intracellular space by coupling to sodium ions (Na^+^) and hydrogen ions (H^+^). ENTs are bidirectional and shuttle adenosine across the plasma membrane with its concentration gradient. Created with BioRender.com, accessed on 13 February 2024.

**Figure 3 cells-13-00381-f003:**
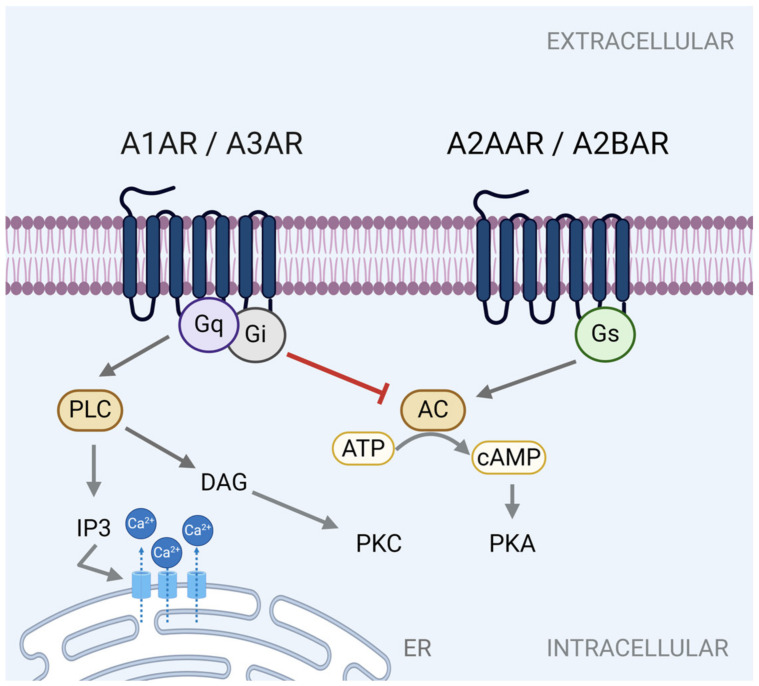
The canonical signaling pathways associated with adenosine receptor activation. Abbreviations: A1 adenosine receptor (A1AR), A2A adenosine receptor (A2AAR), A2B adenosine receptor (A2BAR), A3 adenosine receptor (A3AR), Gs α-subunit (Gs), Gi α-subunit (Gi), Gq α-subunit, adenylyl cyclase (AC), adenosine triphosphate (ATP), cyclic-AMP (cAMP), protein, protein kinase A (PKA), phospholipase C (PLC), inositol 1,4,5-triphosphate (IP3), diacylglycerol (DAG), protein kinase C (PKC), calcium (Ca^2+^), endoplasmic reticulum (ER). Created with BioRender.com, accessed on 13 February 2024.

**Figure 4 cells-13-00381-f004:**
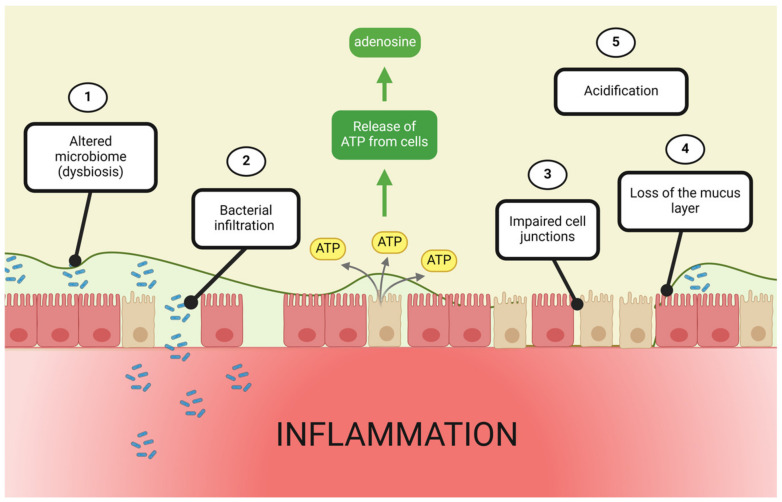
Extracellular adenosine is increased during intestinal inflammation. A key contributing factor to the development of intestinal inflammation is intestinal barrier dysfunction. Barrier dysfunction is associated with an altered microbiome composition (1), bacterial infiltration (2), impaired cell junctions (3), loss of the mucus layer (4) and acidification (5). Collectively, these induce and exacerbate intestinal inflammation. This contributes to the release of ATP from injured cells, which is converted to the alarm molecule adenosine. Created with BioRender.com, accessed on 13 February 2024.

**Figure 5 cells-13-00381-f005:**
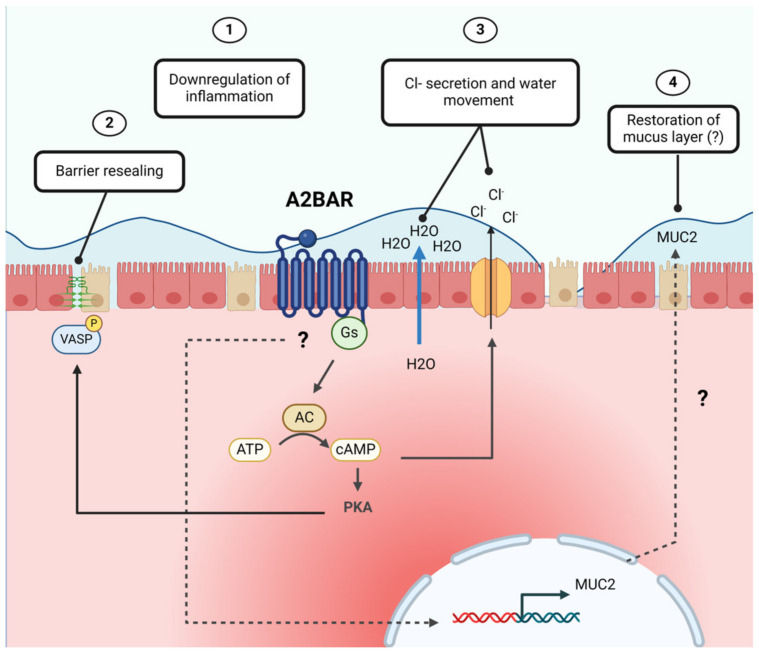
The protective effects of A2BAR signaling on intestinal barrier function. When activated by extracellular adenosine, A2BAR promotes potent downregulation of inflammation (1) and barrier resealing through PKA-phospho-VASP signaling (2). This is accompanied by cAMP-dependent Cl^−^ secretion from intestinal epithelial cells and water movement into the intestinal lumen, which may contribute to mucus hydration and pathogen flushing (3). Evidence also exists to suggest A2BAR signaling may restore the mucus layer by upregulating *Muc2* expression (4). Abbreviations: A2B adenosine receptor (A2BAR), Gs α-subunit (Gs), adenylyl cyclase (AC), adenosine triphosphate (ATP), cyclic-AMP (cAMP), protein, protein kinase A (PKA), vasodilator-stimulated phosphoprotein (VASP), water (H_2_O), chloride ion (Cl^−^), mucin 2 (MUC2). Created with BioRender.com, accessed on 13 February 2024.

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
