# Peer review of "Adenosine in Intestinal Epithelial Barrier Function"

_cells, 2024, doi:10.3390/cells13050381_

Round 1

Reviewer 1 Report

Comments and Suggestions for Authors

The manuscript "Adenosine in intestinal epithelial barrier function" deals with an interesting topic. The review is well organized and writed. The authors have well organized the liretature underlying the importance of adenosine in the gut homeostasis also showing the luck in the literature.

However some minor recvision are nedeed. 

Specifically:

- in the paragraph 2.4 the authors can link the singnaling pathways with a general consequent effect on intesinal epithelial cells functions. 

-the autors have to check all the abbreviations and acronimus

- some spelling errors are present

Author Response

Manuscript cells-2814699

Response to Reviewers

 We would like to extend our sincere thank you to the reviewers for their positive feedback on our manuscript “Adenosine in intestinal epithelial barrier function” and the importance of our review to the field of gut homeostasis. We would also like to thank the reviewers for taking the time to provide thorough and insightful comments on how to improve our manuscript. We have incorporated these suggestions into a revised version of the manuscript. The corresponding revisions appear in red font in the re-submitted manuscript file. In addition, please see below, in blue, a point-by point response to reviewer comments.

Of note:

  • Figure 1 from the original manuscript is now represent by Figure 1, Figure 2 and Figure 3 in the revised version of the manuscript, with minor changes.
  • Section 9: “Conclusions and Future Perspectives” has been re-written to provide a more satisfactory conclusion that addresses the future perspectives, limitations and the therapeutic potential of adenosine, as per reviewer suggestion.

Reviewers' Comments to the Authors

Reviewer 1

  1. Specifically: in the paragraph 2.4 the authors can link the singnaling pathways with a general consequent effect on intesinal epithelial cells functions. 

Author response: We thank the reviewer for this suggestion. We agree it is important to identify the complete signalling pathways responsible for adenosine-mediated effects. Since the downstream pathways can also be triggered by other receptor-mediated events, beyond adenosine, and given that these result in a multitude of possible effects, we have decided to solely focus on the known pathways that drive adenosine-mediated effects in the functional aspects we discuss. As an example in section 5.3, we discuss the impact of A2B adenosine receptor activation, which is tied directly to improved barrier function via activation of protein kinase A(PKA) and PKA-dependent phosphorylation of VASP. We have amended our discussion in Section 2.4 to clarify that this review will expand on the known signalling pathways in subsequent sections an in the context of their direct impact on intestinal epithelial cell function. In our review of the studies undertaken on adenosine signaling, many studies did not identify a specific signaling pathway downstream of receptor activation which we now mention as a limitation our re-written Section 9.

  1. The autors have to check all the abbreviations and acronimus

Author response: We appreciate the reviewers’ attention to this detail and have reviewed all abbreviations and acronyms. The following changes have been made.

  • The acronym ATP was expanded to adenosine triphosphate the first time it is mentioned.
  • The acronym ADP was expanded to adenosine diphosphate the first time it is mentioned.
  • The acronym AMP was expanded to adenosine monophosphate the first time it is mentioned.
  • G-protein coupled receptors previously appeared without an associated acronym. In the revised manuscript the acronym GPCR is added after the first mention of G-protein coupled receptors.
  • Adenylyl cyclase and cyclic AMP previously appeared without an associated acronym. In the revised manuscript the acronym AC is added for adenylyl cyclase and cAMP for cyclic AMP. The sentence in which the two acronyms appear was restructured to accommodate the change.
  • Clarification was provided that Ca2+ refers to calcium.
  • Clarification was provided that COPD refers to chronic obstructive pulmonary disease.
  • Clarification was provided that PPARγ stands for peroxisome proliferator-activated receptor-γ.
  • Tight junctions previously appeared without an associated acronym. In the revised manuscript the acronym TJs is added.
  • Clarification was provided that ZO-1 stands for zonula occludens-1
  1. Some spelling errors are present

Author response: We have reviewed the document and corrected all syntax, grammar and spelling errors. To the best of our knowledge, we can’t identify any further language corrections. 

Reviewer 2 Report

Comments and Suggestions for Authors

Dear Author

This article reviewed the role of adenosine in intestinal epithelial barrier function, which is an important topic. However, some concerns in this article need to be addressed.

1.         Please emphasize the novelty of this review

2.         Please remove “reviewed in:” in lines 26 and 40.

3.         Please write the gene name in italics, e.g. Muc2.

4.         Please indicate the meaning of the abbreviations under the figures.

5.         In line 237, please indicate the TNF type.

6.         In the first citation, please place the full name before the word abbreviation, followed by the abbreviation (e.g. A2BAR, ZO-1, TJ). Please correct.

7.         Please write the name of bacteria such as Salmonella Typhimurium and Salmonella in italics.

8.         Please change "Cl- " to "Cl".

9.         I would suggest including some data on the role of bacteria in adenosine-mediated barrier protection in a table.

10.       Please rewrite and complete the conclusions section.

11.       The limitations of the study were not addressed. Please add them.

Comments on the Quality of English Language

Minor editing of English language required.

Author Response

Manuscript cells-2814699

Response to Reviewers

We would like to extend our sincere thank you to the reviewers for their positive feedback on our manuscript “Adenosine in intestinal epithelial barrier function” and the importance of our review to the field of gut homeostasis. We would also like to thank the reviewers for taking the time to provide thorough and insightful comments on how to improve our manuscript. We have incorporated these suggestions into a revised version of the manuscript. The corresponding revisions appear in red font in the re-submitted manuscript file. In addition, please see below, in blue, a point-by point response to reviewer comments.

Of note:

  • Figure 1 from the original manuscript is now represent by Figure 1, Figure 2 and Figure 3 in the revised version of the manuscript, with minor changes.
  • Section 9: “Conclusions and Future Perspectives” has been re-written to provide a more satisfactory conclusion that addresses the future perspectives, limitations and the therapeutic potential of adenosine, as per reviewer suggestion.

Reviewer 2

  1. Please emphasize the novelty of this review

Author response: We agree with the reviewer on the importance of addressing the novelty of our review and have added an additional paragraph in the introduction section of the revised manuscript addressing this.

  1. Please remove “reviewed in:” in lines 26 and 40.

Author response: We thank the reviewer for the suggestion. We aimed to highlight these specific references as reviews to encourage readers to familiarise themselves on the broader aspects of the topics we are discussing. We will confirm with the editorial team on the standard practice of this journal.

  1. Please write the gene name in italics, e.g. Muc2.

Author response: We have incorporated this suggestion and where the review refers to a gene or an increase in gene expression, we have italicized the gene symbol.

  1. Please indicate the meaning of the abbreviations under the figures.

Author response:  We have incorporated this suggestion and added the meaning of the abbreviations under all of the figures.

  1. In line 237, please indicate the TNF type.

Author response: We thank the reviewer for this observation, and we have clarified that TNF refers to TNF- α, as indicated in the discussion in the original article.

  1. In the first citation, please place the full name before the word abbreviation, followed by the abbreviation (e.g. A2BAR, ZO-1, TJ). Please correct.

Author response: We appreciate the reviewers’ attention to this detail and have reviewed all abbreviations and acronyms. The following changes have been made.

  • The acronym ATP was expanded to adenosine triphosphate the first time it is mentioned.
  • The acronym ADP was expanded to adenosine diphosphate the first time it is mentioned.
  • The acronym AMP was expanded to adenosine monophosphate the first time it is mentioned.
  • G-protein coupled receptors previously appeared without an associated acronym. In the revised manuscript the acronym GPCR is added after the first mention of G-protein coupled receptors.
  • Adenylyl cyclase and cyclic AMP previously appeared without an associated acronym. In the revised manuscript the acronym AC is added for adenylyl cyclase and cAMP for cyclic AMP. The sentence in which the two acronyms appear was restructured to accommodate the change.
  • Clarification was provided that Ca2+ refers to calcium.
  • Clarification was provided that COPD refers to chronic obstructive pulmonary disease.
  • Clarification was provided that PPARγ stands for peroxisome proliferator-activated receptor-γ.
  • Tight junctions previously appeared without an associated acronym. In the revised manuscript the acronym TJs is added.
  • Clarification was provided that ZO-1 stands for zonula occludens-1
  1. Please write the name of bacteria such as Salmonella Typhimurium and Salmonella in italics.

Author response:  We thank the reviewer for this observation and have italicized the names of bacteria in the revised version of the manuscript.

  1. Please change "Cl- " to "Cl".

Author response: We have reviewed the manuscript and we have noted that we reference several other ions such as sodium (Na2+), Hydrogen (H+) and bicarbonate (HCO3-). For these ions we indicate the charge. For consistency we are indicating the charge of chloride as Cl- . However, we agree that [Cl-] can be mistaken for a hyphen and have corrected the manuscript by superscripting all charges to avoid confusion.

  1. I would suggest including some data on the role of bacteria in adenosine-mediated barrier protection in a table.

Author response: We agree with the reviewer and have introduced a table that summarises the relationship between bacteria and adenosine signalling in a table (Table 1).

  1. Please rewrite and complete the conclusions section.

Author response: In the revised version of the manuscript, we have re written Section 9: Concluding remarks and Future Perspectives, where we now discuss adenosine’s potential as a therapeutic target and expand on the unanswered questions related to adenosine’s regulation of barrier function.

  1. The limitations of the study were not addressed. Please add them.

Author response: In the revised version of the manuscript, we have re written Section 9: Concluding remarks and Future Perspectives and addressed the limitation of the literature.

  1. Comments on the Quality of English Language: Minor editing of English language required.

Author response: We have reviewed the document and corrected all syntax, grammar and spelling errors. To the best of our knowledge, we can’t identify any further corrections. 

Reviewer 3 Report

Comments and Suggestions for Authors

The manuscript entitled "Adenosine in intestinal epithelial barrier function" by Stepanova and Aherne is an overview of the role of adenosine in the intestinal epithelium. The manuscript is well written and organized.

However, I have one comment:

-Besides figures some tables could hepl the reader summarizing some aspects reported

Author Response

Manuscript cells-2814699

Response to Reviewers

We would like to extend our sincere thank you to the reviewers for their positive feedback on our manuscript “Adenosine in intestinal epithelial barrier function” and the importance of our review to the field of gut homeostasis. We would also like to thank the reviewers for taking the time to provide thorough and insightful comments on how to improve our manuscript. We have incorporated these suggestions into a revised version of the manuscript. The corresponding revisions appear in red font in the re-submitted manuscript file. In addition, please see below, in blue, a point-by point response to reviewer comments.

Of note:

  • Figure 1 from the original manuscript is now represent by Figure 1, Figure 2 and Figure 3 in the revised version of the manuscript, with minor changes.
  • Section 9: “Conclusions and Future Perspectives” has been re-written to provide a more satisfactory conclusion that addresses the future perspectives, limitations and the therapeutic potential of adenosine, as per reviewer suggestion.

Reviewer 3

  1. Besides figures some tables could hepl the reader summarizing some aspects reported

Author response: We agree with the reviewer and have introduced a table that summarises the relationship between bacteria and adenosine signalling in a table (Table 1).

Reviewer 4 Report

Comments and Suggestions for Authors

In this article, authors have reviewed various aspects of adenosine after inflammation. In my opinion this article merits publication provided they address following minor comments.

·        Number of figures are only three. They can be increased.

·        Not a single table is present in review article. Authors can summarize important findings of subsection in tabulated form.

·        What is Tj in section 5.1? Tight junction or something else.

·        What are potential future prospective? May be a section before conclusion highlighting adenosine or related compounds and receptors for diseases treatment would be interesting for reader.

Comments on the Quality of English Language

Minor editing

Author Response

Manuscript cells-2814699

Response to Reviewers

We would like to extend our sincere thank you to the reviewers for their positive feedback on our manuscript “Adenosine in intestinal epithelial barrier function” and the importance of our review to the field of gut homeostasis. We would also like to thank the reviewers for taking the time to provide thorough and insightful comments on how to improve our manuscript. We have incorporated these suggestions into a revised version of the manuscript. The corresponding revisions appear in red font in the re-submitted manuscript file. In addition, please see below, in blue, a point-by point response to reviewer comments.

Of note:

  • Figure 1 from the original manuscript is now represent by Figure 1, Figure 2 and Figure 3 in the revised version of the manuscript, with minor changes.
  • Section 9: “Conclusions and Future Perspectives” has been re-written to provide a more satisfactory conclusion that addresses the future perspectives, limitations and the therapeutic potential of adenosine, as per reviewer suggestion.

Reviewer 4

  1. Number of figures are only three. They can be increased.

Author response: We have incorporated this suggestion and now present five figures in the revised version. Figure 1 focuses on the mechanisms of adenosine production. Figure 2 focuses on the regulation of adenosine across the plasma membrane. Figure 3 focuses on adenosine receptor signalling. This changed from the original Figure 1 was made to enhance legibility and reader comprehension and has resulted in more focused figures that relate to each section.

  1. Not a single table is present in review article. Authors can summarize important findings of subsection in tabulated form.

Author response: We agree with the reviewer and have introduced a table that summarises the relationship between bacteria and adenosine signalling in a table (Please see Table 1)

  1. What is Tj in section 5.1? Tight junction or something else.

Author response: We thank the reviewer for pointing this out. Tight junctions previously appeared without an associated acronym, and we have corrected this in the revised manuscript and clarified that TJ refers to tight junction the first time it appears in the text.

  1. What are potential future prospective? May be a section before conclusion highlighting adenosine or related compounds and receptors for diseases treatment would be interesting for reader.

Author response: We agree with the reviewer and have re-written the conclusion section. The conclusion section, now titled “Concluding remarks and Future Perspectives” discusses the therapeutic potential of adenosine and emerging therapeutic strategies. In addition we have highlighted key areas in the filed that require further research to enhance our understanding of adenosine-mediated barrier protection.

  1. Comments on the Quality of English Language Minor editing

Author response: We have reviewed the document and corrected all syntax, grammar and spelling errors. To the best of our knowledge, we can’t identify any further corrections. 

We would like to thank the reviewers for their time to view our responses and we look forward to hearing any additional feedback on our revised version of the manuscript.